# The Influence of Maternal Condition on Fetal Cardiac Function during the Second Trimester

**DOI:** 10.3390/diagnostics13172755

**Published:** 2023-08-25

**Authors:** Shifa Yao, Tian Yang, Xiaoxiao Kong, Yuanyuan Dang, Ping Chen, Mingli Lyu

**Affiliations:** 1Ultrasound Department, The International Peace Maternity & Child Health Hospital, School of Medicine, Shanghai Jiao Tong University, Shanghai 200030, China; 18601798023@163.com (S.Y.); shuangyu1201@163.com (T.Y.); 13591820676@163.com (X.K.); yuanyuandang@sohu.com (Y.D.); gfychaoshengke@163.com (P.C.); 2Shanghai Key Laboratory of Embryo Original Diseases, Shanghai 200030, China

**Keywords:** gestational hypertension, gestational diabetic mellitus, maternal age, body mass index, fetal cardiac function

## Abstract

Objective: Maternal health has a direct, profound and lasting effect on the formation and development of the fetal cardiovascular system. The aim of this research was to find whether maternal age, BMI hypertension (GH) or gestational diabetic mellitus (GDM) would affect fetal cardiac function in the second trimester. Method: 329 mothers who had a fetal echocardiogram examination at the International Peace Maternity & Child Health Hospital of China Welfare Institute, Shanghai, China, from 1 January 2020 to 30 April 2020 were enrolled at the gestational age of 21 to 26 weeks (mean 22.78 ± 1.13 weeks). Single-factor analysis and multi-factor line regression analysis were used to find the contribution values of each factor to fetal cardiac function. Results: at the second trimester, maternal age had a minor influence on the fetal left ventricle diastolic function. Higher maternal BMI could cause a decrease in the fetal diastolic function of both the left and right ventricle and the systolic function of the left ventricle. Maternal hypertension and gestational diabetic mellitus had a profound influence on both the left and right fetal heart ventricles of both systolic and diastolic function. Conclusion: maternal condition will have a profound influence on fetal cardiac function as early as the second trimester.

## 1. Introduction

Early pregnancy is a sensitive period for fetal cardiac development, and by the end of the first trimester, the fetal cardiac system is usually largely developed and functional. Therefore, maternal health prior to or in the early stage of pregnancy has a direct effect on the development of fetal cardiac structures and functions, and may also have a lasting effect on the offspring even in adulthood. As early as 1990, Ishii et al. proposed that early life factors may permanently affect cardiac growth and development because the human heart has its highest growth rates in fetal and early postnatal life [1]. Obviously, maternal health before and during pregnancy is most critical for fetal environment exposures and has immediate and long-term effects on offspring cardiac growth and development [2]. An increasing number of maternal-related factors, such as febrile, infectious or chronic diseases, exposure to teratogens, consumption of alcohol and drugs and obesity, are viewed as being closely associated with congenital heart disease (CHD). However, there are also many factors that may not be severe enough to lead to CHD but could cause impaired fetal cardiac function. For example, functional cardiac impairment in the fetuses of mothers with gestational diabetes mellitus (GDM) could occur as early as 13 gestational weeks [3].

Fetal echocardiography involving views of the four chambers, left and right outflow tracts and the three vessels and trachea (3VT) are recommended in abnormal scans. However, fetal cardiac function evaluation is not a routine examination item and may only be performed in fetuses with severe CHD. Therefore, although the risk factors of CHD have been well studied, little is known about fetal cardiac function and its influencing factors.

This study aimed to assess fetal cardiac function, including systolic, diastolic and global myocardial performance in the second trimester, and then to analyze the relationships between functional change and maternal age, body mass index (BMI), gestational hypertension (GH) and GDM.

## 2. Materials and Methods

This study is a prospective cross-sectional investigation that recruited mothers in the second trimester and followed up until their babies were born. All women presented at the antenatal clinic of the International Peace Maternity & Child Health Hospital of China Welfare Institute, Shanghai, China, from 1 January 2020 to 30 April 2020 were enrolled. Exclusion criteria include the presence of structural abnormalities in the fetal heart and in other parts of the body, fetal arrhythmias, twin or multiple pregnancies, intrauterine growth restrictions (defined as a weight for gestation lower than the 5th percentile), macrosomia, abnormal thyroid function of mother and the fetus from in vitro fertilization.

The echocardiographic machine PHILIPS IE33 Ultrasound System (Philips, Ultrasound, 22100 Bothell Everett Highway, Bothell, WA, USA) with the curvilinear 4C probe and standard fetal software was used to perform fetal cardiac structure scanning and functional analysis. All the examination steps were performed in accordance with the recommendations of the American Society of Echocardiography and the China Society of Echocardiography. A standard apical four-chamber view was used for pulsed Doppler interrogation, with the sampling volume placed just distal to the valve tips. To obtain more accurate measurements, we performed the examination when the fetus was in an optimal position and was relatively quiet. The sample volume was as vertical as possible with respect to the cardiac apex, and the vector was arranged to be as parallel as possible to the valve. The sample volume was placed 2 mm above the mitral or tricuspid valve tip.

The peak value of the early diastolic waves (E) of the mitral valve (MVE) and tricuspid valve (TVE) and the peak value of the late diastolic waves (A) of the mitral valve (MVA) and tricuspid valve (TVA) were recorded from 3 consecutive cardiac cycles displaying the highest measurable velocity profiles. The ratio of the peak velocities of the E to A waves (E/A ratio) at both the mitral and tricuspid valves was also obtained. The time intervals from the opening to the closure of the mitral valve (LVA) and tricuspid valve (RVA) and the time interval from the opening to the closure of the aortic valve (LVB) and pulmonary valve (RVB) were recorded. The Tei index was calculated by the following formula: left Tei = (LVA − VB)/LVB; and right Tei = (RVA − RVB)/RVB. E-wave velocity represents passive ventricular filling; A-wave velocity depicts active ventricular filling due to atrial contraction; E/A represents ventricular filling pressures or ventricular compliance; and Tei represents the global function. All these measurements were performed according to standard techniques described in other papers [3,4].

When analyzing the influence factors of maternal health on fetal cardiac function, we first selected one factor at one time and then used multi-factor line regression analysis to find contribution values of each factor to fetal cardiac function.


**Age**


Maternal age was defined as the age of the mother at the time of the fetal cardiac examination. We divided all patients into four cohorts: ≤30 years old (n = 137), 31 to 35 years old (n = 115), 35 to 40 years old (n = 57), and >40 years old (n = 15). One-way analysis of variance (ANOVA) was used for statistics.


**BMI**


Maternal BMI was calculated by the following formula: BMI = weight (kg)/height (m)^2^, with both weight and height measured at the time of the clinical confirmation of pregnancy, which usually occurs at a very early stage in the first trimester (before 6 gestational weeks). We divided all patients into the following three groups according to their BMI and based on recommendations by WHO experts for Asians: ≤24, >24 and ≤28 and >28 [5].


**GDM**


The diagnosis criteria of GDM are as follows. For patients at 24 to 28 gestational weeks, if maternal blood glucose levels are higher than 5.1 mmol/L in fasting blood glucose, or more than 10.0 mmol/L at 1 h oral glucose tolerance test (OGTT), or more than 8.5 mmol/L at 2 h OGTT, a diagnosis of GDM will be established. For patients under 24 gestational weeks, all are given blood glucose testing at their first visit to the doctor for pregnancy. If the maternal blood glucose levels are greater than 7.0 mmol/L for fasting blood glucose, or greater than 6.5% for glycemic control glycollate hemoglobin (HbA1C), or greater than 11.1 mmol/L at 2h OGTT for blood glucose levels, or greater than 11.1 mmol/L for random blood sugar levels, a diagnosis of GDM will be established. All patients diagnosed with GDM were treated with diet control only or diet control accompanied with pharmacological therapy (insulin or metformin), and the GDM was well controlled at the time of fetal cardiac scanning.

The differences in fetal cardiac function in the GDM and non-GDM groups were analyzed using independent samples *t*-tests.


**GH**


Maternal blood pressure was measured at the antenatal clinic using single brachial blood pressure with the patient in a seated position with the appropriate cuff size. All the mothers were offered a blood pressure test every time they presented at the antenatal clinic. If the indexes of blood pressure at two different times were both more than 140/90 mmHg (systolic BP ≥ 140 mmHg and/or diastolic BP ≥ 90 mmHg), the diagnosis of GH was established, according to the International Society for the Study of Hypertension in Pregnancy. All patients diagnosed with GH needed to consent to pharmacological therapy and monitoring. Blood pressure was well controlled when fetal cardiac was examined.

The differences in fetal cardiac function between the GH and non-GH groups were analyzed using independent samples *t*-tests.


**Statistical analysis**


SPSS 24.0 was used for all data analyses. Quantitative analyses were reported as the mean ± standard deviation (SD), and an independent samples *t*-test (two groups) or ANOVA (more than two groups) was used for each variable. Qualitative analyses were reported as number (percentage), and chi-square tests of the observed frequencies were used for each variable. Multi-factor linear regression analysis was used to analyze every factor’s influence on each cardiac function index. A *p* < 0.05 was considered statistically significant.

## 3. Results


**The study included 329 mothers who had a fetal echocardiogram examination and were enrolled at a gestational age of 21 to 26 weeks (mean 22.78 ± 1.13 weeks).**


### 3.1. Single-Factor Analysis

#### 3.1.1. Age (Table 1)

There were no significant differences in gestational weeks among the four age groups. The mean BMI and number of patients with GH or GDM were significantly different among the four age groups. The >40-year-old age group had the highest BMI (25.64 ± 2.61), most of these patients had GH (23.08%, 6/26) and GDM (34.62%, 9/26). The cardiac function results for all patients showed that fetal LVA, RVA and left Tei and right Tei increased with increasing maternal age, especially in patients older than 40 years old. And from the formula for Tei, we could see that an increase in LVA and RVA lead to an increase in left Tei and right Tei. It means that the diastolic function of both left and right ventricles decreases in fetuses with older mothers.

**Table 1 diagnostics-13-02755-t001:** Comparison of maternal demography data and fetal cardiac function in different age groups.

	≤30	>31 and ≤35	>35 and ≤40	>40	*p*
Number	137	115	60.00	26	
Gestation Week	22.62 ± 1.00	22.82 ± 1.28	22.95 ± 1.11	23.2 ± 0.95	0.08
BMI (kg/m^2^)	23.94 ± 2.88	24.62 ± 2.97	24.66 ± 3.05	25.64 ± 2.61	0.03
GH	8 (5.84%)	11 (9.57%)	11 (19.30%)	6 (23.08%)	0.01
GMD	11 (8.03%)	18 (15.65%)	16 (28.07%)	9 (34.62%)	0.00
LVEF (%)	67.24 ± 3.74	67.92 ± 3.25	67.28 ± 3.76	68.39 ± 3.43	0.26
MVE (cm/s)	30.73 ± 4.16	30.03 ± 4.16	30.04 ± 5.06	29.34 ± 4.26	0.41
MVA (cm/s)	46.53 ± 6.61	45.61 ± 6.04	46.36 ± 6.00	46.98 ± 5.79	0.61
LVA (ms)	250.50 ± 9.85	253.68 ± 8.45	254.27 ± 8.68	255.44 ± 8.91	0.02
LVB (ms)	175.98 ± 7.39	174.41 ± 8.27	174.80 ± 7.10	176.11 ± 9.37	0.53
TVE (cm/s)	33.83 ± 4.70	33.55 ± 5.39	33.46 ± 5.49	32.20 ± 5.23	0.53
TVA (cm/s)	50.57 ± 7.0	50.17 ± 6.42	50.67 ± 6.34	50.96 ± 7.37	0.93
RVA (ms)	256.86 ± 11.46	258.11 ± 1.95	260.31 ± 44	261.72 ± 12.23	0.04
RVB (ms)	183.77 ± 7.27	182.89 ± 7.17	181.07 ± 7.21	179.33 ± 7.77	0.06
Left Tei	0.43 ± 0.05	0.44 ± 0.07	0.45 ± 0.05	0.45 ± 0.06	0.02
Right Tei	0.40 ± 0.05	0.43 ± 0.06	0.43 ± 0.06	0.44 ± 0.06	0.00
MVE/A	0.66 ± 0.06	0.66 ± 0.08	0.66 ± 0.09	0.63 ± 0.07	0.11
TVE/A	0.67 ± 0.06	0.66 ± 0.06	0.66 ± 0.07	0.65 ± 0.06	0.42

#### 3.1.2. BMI (Table 2)

For all patients, there were no significant differences in the patient’s age or gestational week in the different BMI groups. In the >28 BMI group, most of them had GH (25.00%, 10/40) and GDM (32.50%, 13/40) compared to the other two groups (*p* = 0.07, *p* = 0.00). The results showed that the value of the RVA and left Tei and right Tei increased; MVE, TVE, LVB, MVE/A and TVE/A decreased, whereas the BMI increased. It means that the diastolic function of both the left and right ventricles and the systolic function of the left ventricle were decreased in fetuses with higher BMI mothers.

**Table 2 diagnostics-13-02755-t002:** Comparison of maternal demography data and fetal cardiac function in different BMI groups.

BMI Group (kg/m^2^)	≤24	>24 and ≤28	>28	*p*
Number	156	142	40	
Age	32.04 ± 4.45	32.66 ± 4.78	33.73 ± 4.92	0.11
Gestation week	22.63 ± 1.04	23.00 ± 1.38	22.89 ± 1.14	0.07
Gestation hypertension	7 (4.49%)	19 (13.38%)	10 (25.00%)	0.00
GDM	20 (12.82%)	21 (14.79%)	13 (32.50%)	0.01
LVEF (%)	67.54 ± 3.75	67.30 ± 3.43	68.60 ± 3.23	0.13
MVE (cm/s)	31.36 ± 4.49	30.11 ± 4.10	27.03 ± 3.97	0.00
MVA (cm/s)	46.85 ± 6.72	45.82 ± 5.55	45.19 ± 6.53	0.20
LVA (ms)	252.25 ± 7.58	252.93 ± 9.85	252.48 ± 11.95	0.86
LVB (ms)	177.20 ± 6.67	175.17 ± 7.66	169.45 ± 8.68	0.00
TVE (cm/s)	34.27 ± 5.28	33.67 ± 4.70	30.27 ± 4.95	0.00
TVA (cm/s)	50.95 ± 7.28	50.51 ± 6.16	48.56 ± 5.92	0.13
RVA (ms)	257.88 ± 11.88	259.70 ± 11.61	261.45 ± 12.54	0.01
RVB (ms)	183.85 ± 7.05	182.91 ± 8.04	178.15 ± 7.58	0.27
Left Tei	0.42 ± 0.05	0.44 ± 0.06	0.48 ± 0.07	0.00
Right Tei	0.41 ± 0.05	0.46 ± 0.07	0.46 ± 0.06	0.00
MVE/A	0.67 ± 0.07	0.66 ± 0.07	0.60 ± 0.07	0.00
TVE/A	0.67 ± 0.05	0.66 ± 0.06	0.66 ± 0.06	0.00

#### 3.1.3. GDM (Table 3)

There was no significant difference in gestational week or the number of patients with hypertension between the two groups. The GDM group had older patients (34.43 ± 4.63 years vs. 31.90 ± 4.41 years) and higher BMIs (24.24 ± 2.81 vs. 25.59 ± 3.36) than those of the non-GDM group. MVE, LVB, TVE, RVB, MVE/A and TVE/A decreased, and the left Tei and right Tei index increased in the GDM group compared with those in the non-GDM group. It means that both diastolic and systolic functions in both the left and right ventricles were decreased in fetuses with GDM mothers.

**Table 3 diagnostics-13-02755-t003:** Comparison of maternal demography data and fetal cardiac function in different GDM groups.

	Non-GDM		GDM
Number	278	51	*p*
Age	32.06 ± 4.52	34.81 ± 4.77	0.00
Gestation Week	22.75 ± 1.10	22.94 ± 1.32	0.27
BMI (kg/m^2^)	24.22 ± 2.83	25.52 ± 3.34	0.00
Gestation hypertension	28 (10.07%)	8 (15.69%)	0.28
LVEF (%)	67.62 ± 3.62	67.31 ± 3.28	0.56
MVE (cm/s)	31.15 ± 4.17	25.95 ± 3.32	0.00
MVA (cm/s)	46.31 ± 6.40	45.73 ± 5.39	0.53
LVA (ms)	252.06 ± 8.70	255.75 ± 11.65	0.05
LVB (ms)	176.83 ± 6.87	167.73 ± 7.30	0.00
TVE (cm/s)	34.41 ± 4.91	29.00 ± 3.76	0.00
TVA (cm/s)	50.72 ± 6.90	49.22 ± 5.40	0.14
RVA (ms)	257.85 ± 11.29	265.90 ± 11.86	0.00
RVB (ms)	183.76 ± 7.11	177.12 ± 8.65	0.00
Left Tei	0.43 ± 0.04	0.52 ± 0.06	0.00
Right Tei	0.40 ± 0.05	0.50 ± 0.05	0.00
MVE/A	0.67 ± 0.06	0.57 ± 0.05	0.00
TVE/A	0.68 ± 0.06	0.59 ± 0.05	0.00

#### 3.1.4. GH (Table 4)

There were no significant differences observed between the two groups in gestational weeks and GDM status. Women in the GH group are older (34.81 ± 4.77 vs. 32.06 ± 4.52, *p* < 0.01) and had higher BMIs (26.52 ± 2.73 vs. 26.52 ± 2.73, *p* < 0.01) than those of the non-GH group. Except for LVEF and MVA, all indexes measured were significantly different in the two groups. MVE, LVB, TVE, TVA, MVE/A and TVE/A were significantly lower, and LVA, RVA and left and right Tei indexes were significantly higher in the GH group than in the non-GH group. Thus, maternal GH mainly leads to diastolic dysfunction (represented as E/A) caused by a decrease in the peak value of early diastolic waves (E) and global myocardial performance index (represented as Tei) dysfunction caused by a decrease in ejection time, which also indicates impairments in both left and right ventricles, both diastolic and systolic functions and both local and global myocardial performances.

These results indicate that the fetuses would have different cardiac functions in both the left and right heart ventricles and in global myocardial performance when their mother was older and with higher BMI, GH or GDM.

**Table 4 diagnostics-13-02755-t004:** Comparison of maternal demography data and fetal cardiac function in different gestational hypertension groups.

	Non-Hypertension	Hypertension	*p*
Number	302	36	
Age	32.19 ± 4.52	35.11 ± 5.06	0.00
Gestation week	22.77 ± 1.11	22.89 ± 1.33	0.55
BMI (kg/m^2^)	24.18 ± 2.88	26.52 ± 2.73	0.00
GDM	46 (15.23%)	8 (22.22%)	0.28
LVEF (%)	67.52 ± 3.50	67.97 ± 4.15	0.48
MVE (cm/s)	30.78 ± 4.34	26.44 ± 3.57	0.00
MVA (cm/s)	46.40 ± 6.36	44.75 ± 5.01	0.14
LVA (ms)	251.72 ± 8.97	258.06 ± 9.35	0.00
LVB (ms)	176.22 ± 7.58	169.18 ± 6.15	0.00
TVE (cm/s)	34.11 ± 4.97	28.76 ± 3.90	0.00
TVA (cm/s)	50.77 ± 6.81	48.06 ± 5.17	0.02
RVA (ms)	257.85 ± 11.42	267.53 ± 11.33	0.00
RVB (ms)	183.58 ± 7.30	176.94 ± 8.29	0.00
Left Tei	0.43 ± 0.05	0.53 ± 0.06	0.00
Right Tei	0.41 ± 0.05	0.51 ± 0.04	0.00
MVE/A	0.67 ± 0.07	0.59 ± 0.06	0.00
TVE/A	0.67 ± 0.05	0.60 ± 0.05	0.00

### 3.2. Multi-Factor Linear Regression Analysis (Table 5)

In order to analyze the influence extent of different factors on the fetus’s cardiac function, we used multifactor linear regression analysis (SPSS 24.0). Except for LVEF and MVA, all other items successfully fit regression models which had statistical significance (*p* < 0.05). For MVE, LVB, RVE, RVB, left Tei, right Tei, MVE/A and TVE/A, a mother with GDM was the most important factor in these indexes’ changes. It explained the maximum variation of these indexes. For TVA, the most important factor was BMI. For RVA, the most important factor was GH. At the same time, we realized that maternal age was not the decisive factor for fetal cardiac function, and usually caused a minor change in these indexes.

**Table 5 diagnostics-13-02755-t005:** Multi-factor linear regression analysis of maternal age, BMI, GH and GDM to different fetal cardiac function indexes.

	Beta	SE of Beta	Standardizad Beta	t	*p*	95% CI		Adjusted R Squared	F	*p*
						Upper	Lower			
LVEF										
(Constant quantity)	64.678	2.081		31.075	0.000	60.584	68.772	−0.003	0.712	0.584
Age	0.037	0.044	0.049	0.857	0.392	−0.049	0.123			
GH	0.207	0.659	0.018	0.315	0.753	−1.089	1.504			
GDM	−0.515	0.548	−0.053	−0.939	0.349	−1.594	0.564			
BMI	0.071	0.069	0.059	1.026	0.306	−0.065	0.207			
MVE										
(Constant quantity)	38.251	2.177		17.568	0.000	33.968	42.534	0.300	37.074	0.000
Age	0.053	0.046	0.055	1.150	0.250	−0.037	0.143			
GH	−3.354	0.689	−0.232	−4.866	0.000	−4.710	−1.998			
GDM	−4.730	0.574	−0.388	−8.244	0.000	−5.859	−3.602			
BMI	−0.349	0.072	−0.230	−4.825	0.000	−0.491	−0.207			
MVA										
(Constant quantity)	53.954	3.608		14.953	0.000	46.856	61.052	0.014	2.231	0.065
Age	−0.009	0.076	−0.007	−0.120	0.904	−0.158	0.140			
GH	−0.905	1.142	−0.045	−0.793	0.429	−3.152	1.342			
GDM	−0.130	0.951	−0.008	−0.137	0.891	−2.000	1.741			
BMI	−0.300	0.120	−0.142	−2.500	0.013	−0.535	−0.064			
LVB										
(Constant quantity)	179.891	4.345		41.401	0.000	171.332	188.450	0.327	31.120	0.000
Age	0.248	0.094	0.145	2.646	0.009	0.063	0.433			
GH	−6.858	1.241	−0.303	−5.527	0.000	−9.302	−4.414			
GDM	−9.288	1.124	−0.444	−8.267	0.000	−11.502	−7.075			
BMI	−0.414	0.143	−0.159	−2.895	0.004	−0.696	−0.132			
TVE										
(Constant quantity)	39.834	2.576		15.465	0.000	34.767	44.900	0.258	30.314	0.000
Age	0.066	0.054	0.060	1.215	0.225	−0.041	0.172			
GH	−4.516	0.815	−0.272	−5.539	0.000	−6.120	−2.912			
GDM	−4.987	0.679	−0.356	−7.347	0.000	−6.322	−3.651			
BMI	−0.293	0.086	−0.168	−3.420	0.001	−0.461	−0.124			
TVA										
(Constant quantity)	56.998	3.844		14.826	0.000	49.436	64.561	0.028	3.393	0.010
Age	0.049	0.081	0.034	0.608	0.544	−0.110	0.208			
GH	−2.044	1.217	−0.094	−1.679	0.094	−4.438	0.350			
GDM	−1.091	1.013	−0.060	−1.077	0.282	−3.084	0.901			
BMI	−0.316	0.128	−0.139	−2.474	0.014	−0.567	−0.065			
RVA										
(Constant quantity)	253.232	7.544		33.566	0.000	238.372	268.092	0.127	10.060	0.000
Age	0.014	0.163	0.005	0.087	0.931	−0.307	0.335			
GH	9.285	2.154	0.269	4.310	0.000	5.041	13.529			
GDM	7.735	1.951	0.242	3.965	0.000	3.893	11.578			
BMI	0.119	0.248	0.030	0.480	0.632	−0.370	0.608			
RVB										
(Constant quantity)	189.268	4.802		39.412	0.000	179.809	198.728	0.175	14.146	0.000
Age	−0.066	0.104	−0.038	−0.634	0.527	−0.270	0.139			
GH	−6.128	1.371	−0.271	−4.468	0.000	−8.829	−3.427			
GDM	−6.277	1.242	−0.300	−5.055	0.000	−8.723	−3.831			
BMI	−0.105	0.158	−0.040	−0.666	0.506	−0.417	0.206			
Left Tei										
(Constant quantity)	0.386	0.021		18.145	0.000	0.344	0.428	0.600	127.122	0.000
Age		0.000	−0.075	−2.083	0.038	−0.002	0.000			
GH	0.089	0.007	0.477	13.238	0.000	0.076	0.102			
GDM	0.087	0.006	0.556	15.605	0.000	0.076	0.099			
BMI	0.002	0.001	0.123	3.397	0.001	0.001	0.004			
Right Tei										
(Constant quantity)	0.346	0.023		15.257	0.000	0.302	0.391	0.572	113.671	0.000
Age	0.000	0.000	0.018	0.495	0.621	−0.001	0.001			
GH	0.095	0.007	0.492	13.206	0.000	0.081	0.109			
GDM	0.082	0.006	0.506	13.754	0.000	0.070	0.094			
BMI	0.002	0.001	0.084	2.255	0.025	0.000	0.003			
MVE/A										
(Constant quantity)	0.721	0.032		22.652	0.000	0.658	0.784	0.404	58.057	0.000
Age	0.001	0.001	0.084	1.918	0.056	0.000	0.003			
GH	−0.063	0.010	−0.275	−0.626	0.000	−0.083	−0.043			
GDM	−0.102	0.008	−0.530	−12.204	0.000	−0.110	−0.086			
BMI	−0.003	0.001	−0.138	−3.140	0.002	−0.005	−0.001			
TVE/A										
(Constant quantity)	0.722	0.031		23.109	0.000	0.660	0.784	0.388	40.250	0.000
Age	0.002	0.001	0.120	2.295	0.023	0.000	0.003			
GH	−0.065	0.009	−0.381	−7.280	0.000	−0.083	−0.047			
GDM	−0.067	0.008	−0.426	−8.316	0.000	−0.083	−0.051			
BMI	−0.004	0.001	−0.187	−3.577	0.000	−0.006	−0.002			

## 4. Discussion

The intrauterine environment is critical for fetal organ development, growth and maturation, including in the fetal heart. Maternal health is the most important factor for intrauterine environment formation and has a direct and lasting effect on the offspring at the fetus, childhood and adolescent stages [1,2]. Recently, some researchers have begun to pay attention to fetal cardiac function analysis and have tried to identify the influencing factors, such as maternal blood glycemic levels and blood pressure. However, the knowledge is still limited and uncomprehensive.

In this study, we evaluated fetal cardiac performance in the second trimester and found that maternal overweight before pregnancy, GDM and GH could severely alter fetal cardiac function, while maternal age could lead to a minor diastolic function alternation.


**Age**


Many studies have shown that maternal age is a risk factor for CHD [6,7]. After cases with chromosomal anomalies were excluded, there was a greater prevalence of CHD in mothers over 35 years old, with effect sizes ranging between 12% and 36% for increased risk. However, the research from Best and Rankin, which included 4024 singleton cases of nonchromosomal CHD, showed little evidence of advanced maternal age and risk factors for CHD [8]. Although CHD leads to changes in fetal cardiac function, we focused on the relationships between maternal age and fetal cardiac function in non-CHD and in structurally normal fetuses.

Our results showed that although some fetal cardiac function indexes were altered in women of advanced maternal age, especially when maternal age was older than 40 years old, the vast majority of these changes were caused by complications related to maternal age because the standardized Beta values were much smaller than those of the other three factors. And it was obvious that there were more patients with GH, GDM and/or higher BMI in the >40-year-old group than in the other three groups. In the single-factor analysis, increasing the time interval from the opening to the closure of the mitral valve (LVA) and tricuspid valve (RVA) lead to an increase in the global functioning in the left and right heart. Increasing the LVA and RVA means they needed more time for the blood to flow through the mitral and tricuspid valves, and the myocardia relaxation may be worse with maternal age. From the results of the multi-factor linear regression analysis, the LVB and left Tei show a positive correlation with maternal age. Combining all the results, we thought that maternal age mainly influences the opening time of the mitral valve and then the fetal left ventricle diastolic function. Though usually, the increasing of Tei means a worse heart function, we do not know if such change is beneficial or harmful to the fetus and maybe it was just an adaptive change.

Gillman et al. found that a higher maternal age is associated with higher newborn systolic blood pressure and may represent similar phenomena in fetal cardiac function [9]. However, the research did not exclude mothers with gestational complications. The research from Cooke et al. revealed that the offspring from rats of an advanced maternal age are at an increased risk of adult cardiovascular disease, associated with endothelium-dependent relaxation and oxidative stress [10]. Although these studies did not suggest that advanced maternal age could directly lead to a worse fetal cardiac function, they indicated that advanced maternal age could cause functional changes in the offspring’s heart. Notably, these studies combined the influence of maternal age with that of maternal GH, GDM, BMI or other confounding factors; this combination is the main differentiating factor between other studies and our study.


**BMI**


With a drastic increase in the prevalence of obesity in China and other countries worldwide, an increasing number of overweight and obese women are becoming pregnant. In our study, women’s BMI had an important influence on fetal cardiac function. The obese women’s group had the most patients with GH and/or GDM. At the same time, all dramatic changes occurred in the obese group (BMI > 28) compared with the other two groups. Higher BMI could cause the diastolic function of both the left and right ventricles and the systolic function of the left ventricle to decrease in the fetal heart. But its influence is still minor, only stronger than maternal age and far less than maternal GH and GDM.

An increasing number of studies have shown that obesity is an important factor associated with pregnancy complications and causes an increased risk of maternal and fetal morbidity and mortality [11]. Evidence has shown that the prevalence of CHD is higher in overweight and obese women [12]. Rat studies and human clinical analyses have shown that maternal BMI is associated with blood pressure in the offspring [13,14,15]. Ojala et al. collected fetal information at 1 h after membrane rupture in the first phase of labor and found that maternal pre-pregnancy BMI is associated with fetal sympathetic activation, and the latter is considered a risk factor for cardiovascular diseases [16]. Kristin et al. revealed that the fetuses of obese women showed lower heart rate variability, less heart rate acceleration, more vigorous motor activity and lower cardiac-somatic integration than those from normal prepregnant BMI women [17]. Some controversial views also exist, such as the results from Dewi et al., where they thought that maternal obesity had nothing to do with fetal cardiac autonomic nervous system development, even though these patients had higher insulin resistance and inflammation [18]. Geelhoed et al. also found that maternal BMI had no association with longitudinally measured neonatal left ventricular mass (LVM, an index for heart size) from 6 weeks to 6 months [2]. However, most of these studies do not take patient age, GH and GDM status into account and all these factors can affect fetal cardiac function, as revealed in our research. Additionally, these studies focused on offspring after birth and did not focus on alterations at the fetal stage, especially during the second trimester.


**GDM**


In 2004, some scholars revealed that GDM had a close connection with increased congenital malformations, perinatal mortality and morbidity, growth restriction and fetal organ damage [19]. Usually, maternal hyperglycemia leads to a 2- to 5-fold higher risk of CHD [20], which is a genetic diagnosis for only 11% of probands [21]. Poor maternal glycemic control sometimes leads to a very poor prognosis in the fetus due to interventricular septum thickness hypertrophy, transient hypertrophic subaortic stenosis and different degrees of left ventricular outflow obstruction [22,23]. Strict glycemic control could not prevent abnormal cardiac growth, cardiac hypertrophy or diastolic dysfunction in GDM, even at normal fetal weight, regardless of the examination performed in fetuses or neonates [19,24]. Mdaki et al. designed a mouse study to investigate the effect of a maternal high-fat diet on fetal cardiac development, independent of GDM. They found that infants from diabetic rats with a high-fat diet had more severe cardiac dysfunction, similar to that in adult diabetic cardiomyopathy [25]. Sanhal et al. revealed that fetal left ventricular systolic and diastolic dysfunctions in GDM or pre-GDM mothers were worse than that in fetuses from normal mothers [26]. Even as early as the first trimester, maternal hyperglycemia, no matter diagnosed with pregestational diabetes or GDM in the second and third trimester, could lead to an increase in the fetal heart rate [27,28]. The mechanism may be that glucose passes the placenta into the fetus and increases fetal insulin production. Insulin is the main fetal growth factor and may stimulate fetal body and heart growth [18]. During pregnancy, cardiomyocytes shift their major energy source from glucose in the early developmental stages to fatty acid oxidation shortly after birth to meet the high energy demand of the maturing heart. Most scholars thought that the fuel type used by the cells serves not only as a source of energy but also as a critical regulator of self-renewal and the differentiation of stem or progenitor cells [29]. Using human embryonic stem cell-derived cardiomyocytes (hESC-CMs), Haruko N et al. verified that high glucose inhibited the maturation of cardiomyocytes at the genetic, structural, metabolic, electrophysiologic and biomechanical levels by promoting nucleotide biosynthesis through the pentose phosphate pathway, which showed increased mitotic activity and decreased maturity [30]. Luckily, Hornberger et al. found that most affected infants are clinically asymptomatic, and cardiac function would improve in months [31].

In our research, we found that both diastolic dysfunction and general cardiac performance in the left and right ventricles decreased in the fetuses of GDM women, even when maternal blood glucose is well controlled. At the same time, maternal GDM is the most important factor in fetal cardiac function compared with the other three factors, except for TVA and RVA. These results reveal that GDM, even well controlled, could directly impair fetal cardiac function. Our results are consistent with those of the above-mentioned studies. Additionally, a study from Atiq et al. on the fetal cardiac function of mothers with GDM during the second trimester showed the following: (1) in the fetuses of diabetic mothers, isovolumetric relaxation and contraction times were significantly prolonged; (2) myocardial performance indexes (MPI, also known as Tei) in both the left and right ventricles significantly increased; and (3) mitral E/A ratios significantly decreased [32]. These results are consistent with ours, except that the tricuspid valve E/A ratio was not different between the two groups. The different exclusion criteria in the two studies represent the main reason for this difference. A systematic review and meta-analysis study from Sirico A et al. which evaluated fetal MPI in the third trimester of pregnant women with GDM also demonstrated an increase in the fetal left MPI [33]. All these results showed that maternal hyperglycemia will have a profound and lasting effect on fetal cardiac function.


**GH**


Many studies have shown that offspring exposed to maternal preeclampsia have an increased risk of higher blood pressure, BMI, cardiovascular diseases and stroke during childhood and early adolescence [34,35]. Some animal studies have shown that altered pressure loads and hypoxia during fetal myocardium development might have a long-lasting effect on its structure and function [36,37]. Using data from 1592 patients, Timpka S et al. showed that fetal exposure to preeclampsia is associated with a more adverse cardiac structure in youth (17 years old) but was not related to cardiac function [38]. These results are consistent with those from another large population study comprising 1006 patients, which found that abnormalities of the retinal microvasculature and cardiac structure are observed in the offspring of women with hypertensive disorder pregnancy in mid-adulthood (mean age 40 years) [39]. All the data listed above are related to the relationship between maternal hypertension and the offspring’s cardiac structural and functional alterations, but the data does not include changes in the fetus.

In our studies, we found that as early as the second trimester, fetal cardiac function showed a worse performance in GH pregnancy. These fetuses showed lower E peak values in early diastolic waves and diastolic function (E/A) in both the mitral and tricuspid valves, shorter ejection times in both the left and right ventricles, and higher total cardiac function (Tei) in both the left and right ventricles compared with fetuses from non-GH mothers. So maternal hypertension could impair both the diastolic and systolic functions of the fetus. We have not found similar studies on the relationship between fetal cardiac function and hypertension, except for a mouse study by Armstrong et al., the findings of which were consistent with ours [40]. They used a mouse model to demonstrate that gestational hypertension could lead to increases in LV Gata4, Gate 6 and P300 in offspring and remodeling of the fetal left ventricular structure and could finally result in fetal diastolic dysfunction.


**Strengths and limitations**


Our research is a cross-sectional case-control study. To have more truthful analysis results, we conducted this study with very strict criteria and used both single-factor analysis and multi-factor linear regression analysis. Despite the procedures adopted to ensure well-controlled and high-quality data, some limitations still need to be considered. The most important limitation of our study is the relatively small sample size. Additionally, the selection bias cannot be ignored. Furthermore, several pregnancy complications have been thought to show a gender-specific pattern [41,42], and we did not take this factor into account. The inheritance background, which most likely plays an important role in the incidence of CHD [43,44], was also not considered. We were unable to follow up with these fetuses and re-evaluate their cardiac function after they were born.

## 5. Conclusions

As shown above, fetal cardiac function is affected by multiple factors. The maternal age has minor influence on fetal left ventricle diastolic function. The diastolic function of both left and right ventricles and the systolic function of the left ventricle were decreased in fetuses with higher BMI mothers. Maternal GDM and GH could lead to fetal cardiac dysfunction in both the left and right ventricles, and the results were consistent with those fetuses analyzed after birth or with animals.

Changes in fetal cardiac function may be a means of fetal self-regulation to a specific environment. A functional assessment of fetal cardiac function is important for monitoring fetal intrauterine status and predicting outcomes, especially for high-risk pregnant women. Further studies are needed to enlarge the sample size and to identify the underlying causal mechanisms and long-term consequences.

## Data Availability

The datasets generated during and/or analyzed during the current study are not publicly available but are available from the corresponding author upon reasonable request.

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
