# Peer review of "The Influence of Maternal Condition on Fetal Cardiac Function during the Second Trimester"

_diagnostics, 2023, doi:10.3390/diagnostics13172755_

Round 1

Reviewer 1 Report

Authors present an interesting study evaluating the fetal cardiac function in fetuses from pregnant women with impaired health conditions during the second trimester. I have some suggestions that could improve the manuscript:

Title: “Maternal Health Station” does not have a clear meaning in English, please correct as “maternal condition”. Use capital letters only at the beginning of the sentence when there are no proper nouns.

Line 42: previous Authors have also demonstrated that pregestational and gestational diabetes are associated with changes in fetal heart rate in the first trimester compared to healthy controls (PMID: 30465928, PMID: 31706055), highlighting the impact of hyperglycemia on fetal heart function already in the first trimester.

Line 54: in the Methods section authors should not include the numbers of included patients as well as mean of gestation weeks or other data that belong to the Results section, they should only include the study design (This study is a prospective cross-sectional investigation including pregnant women undergoing fetal echocardiogram examination between 21 and 26 gestational weeks).

Line 61: fetus

Line 108 and 119: pharmacological therapy

Line 139: Tei instead of Tie?

Line 190: “the” without capital letters

Line 309: Authors results are consistent also with a sytematic review and metanalysis that demonstrated an increase in fetal MPI in diabetic women in the third trimester (PMID: 34425601).

Minor language flaws, please re-check carefully the whole manuscript.

Author Response

Point 1: “Maternal Health Station” does not have a clear meaning in English, please correct as “maternal condition”. Use capital letters only at the beginning of the sentence when there are no proper nouns?

Respond:Thank you for your advice. We correct “Maternal Health Station” as “maternal condition” and correct first word “The” as “the” in the title.

Point 2: previous Authors have also demonstrated that pregestational and gestational diabetes are associated with changes in fetal heart rate in the first trimester compared to healthy controls (PMID: 30465928, PMID: 31706055), highlighting the impact of hyperglycemia on fetal heart function already in the first trimester.

Respond: Thank you for your advice. We learned the two papers and found that they were all about the fetal heart rate and maternal gestational or pregestational diabetes. All these results showed that maternal hyperglycemia could lead increasing of fetal heart rate. So we added these studies in our manuscript in line 287-289.

Point 3: in the Methods section authors should not include the numbers of included patients as well as mean of gestation weeks or other data that belong to the Results section, they should only include the study design (This study is a prospective cross-sectional investigation including pregnant women undergoing fetal echocardiogram examination between 21 and 26 gestational weeks)

Respond:. Thank you for your advice, we moved this section form the Methods section to the Results section.

Point 4: Line 61: fetus

Respond: corrected.

Point 5: Line 108 and 119: pharmacological therapy

Respond: corrected.

Point 6: Line 139: Tei instead of Tie?

Respond: corrected.

Point 7: Line 190: “the” without capital letters

Respond: corrected.

Point 8:Line 309: Authors results are consistent also with a sytematic review and metanalysis that demonstrated an increase in fetal MPI in diabetic women in the third trimester (PMID: 34425601).

Respond: We read the article, and added their results in our manuscript. Thanks for your advice.

Point 9:Comments on the Quality of English Language: Minor language flaws, please re-check carefully the whole manuscript.

Respond:Sorry for the mistakes, we re-checked the manuscript and corrected them.

Reviewer 2 Report

Dear author's

I was pleased to review your article and i have the following comments:

The theme is interesting but there are some recommendation in order to improve methods. 

Title> i recommend you to remove station from the article title.

Methods> The measurement's were preformed by the same doctor?

It is not clear if the study was prospective or retrospective.

Please explain why this measurable items were used (MVE) / (TVE) /(MVA) and (TVA) and  (E/A ratio)?

Please explain the novelty of your research. 

The limitation of the study should be highlighted.

English end punctuation edits.

Author Response

Point 1:  i recommend you to remove station from the article title.

Respond: Thank you for your advice. We correct “Maternal Health Station” as “maternal condition”

Point 2: The measurement's were preformed by the same doctor?

Respond: Yes, all the measurements were preformed by Shifa Y.

Point 3: It is not clear if the study was prospective or retrospective.

Respond: This study was prospective. We evaluated all fetal cardiac function in second trimester, then followed up.

Point 4: Please explain why this measurable items were used (MVE) / (TVE) /(MVA) and (TVA) and  (E/A ratio)?

Respond: Because for adult, these items can also reflect the change of cardiac function. For our research was prospective, we do not know if maternal condition would cause change in fetal cardiac function. Maybe these factors could not cause drastic changes of Tei, but reduce peak flow velocity altered. E/A ratio can reflect diastolic dysfunction.

Point 5: Please explain the novelty of your research. 

Respond: A functional assessment of fetal cardiac function is important for monitoring fetal intrauterine status and predicting outcomes, especially for high-risk pregnant women.our results showed that Maternal health have a direct, profound and lasting effect on the formation and development of fetal cardiovascular system, especially maternal hyperglycemia and hypertension. Next, we plan to evaluate caidiac function for neonatal, one-year old and three-year old child from GHD and GH mother.

Point 6: The limitation of the study should be highlighted.

Respond: The limitation of the study is in the strengths and limitations in line 348-352.

Point 7:English end punctuation edits.

Respond: Sorry for the mistakes, we re-checked the manuscript and corrected them.

Round 2

Reviewer 2 Report

Thank you for your revised version of the manuscript. 
Some punctuation edits should be done to the manuscript.